# Epidemiological characterization of ischemic heart disease at different altitudes: A nationwide population-based analysis from 2011 to 2021 in Ecuador

Esteban Ortiz-Prado[1]*, Juan S. Izquierdo-Condoy[1], Raúl Fernández-Naranjo[1], Jorge Vásconez-González[1], Leonardo Cano[1], Ana Carolina González[1,2], Estefanía Morales-Lapo[1], Galo S. Guerrero-Castillo[1], Erick Duque[1], Maria Gabriela Davila Rosero[1], Diego Egas[3], Ginés Viscor[4]

1 One Health Research Group, Faculty of Medicine, Universidad de las Américas, Quito, Ecuador, 2 Pós Graduação de Clinica Medica, Universidade Federal de Ciências da Saúde de Porto Alegre, Porto Alegre, Brasil, 3 Departamento de Cardiología, Hospital Metropolitano, Quito, Ecuador, 4 Departament de Biología Cel·lular, Fisiologia i Immunologia, Universitat de Barcelona, Barcelona, Spain

* e.ortizprado@gmail.com

**Data Availability Statement:** The data generated and analyzed during this study are available for

## Abstract

### Background

Cardiovascular diseases, including ischemic heart disease, are the leading cause of premature death and disability worldwide. While traditional risk factors such as smoking, obesity, and diabetes have been thoroughly investigated, non-traditional risk factors like high-altitude exposure remain underexplored. This study aims to examine the incidence and mortality rates of ischemic heart disease over the past decade in Ecuador, a country with a diverse altitude profile spanning from 0 to 4,300 meters.

### Methods

We conducted a geographic distribution analysis of ischemic heart disease in Ecuador, utilizing hospital discharge and mortality data from the National Institute of Census and Statistics for the years 2011–2021. Altitude exposure was categorized according to two distinct classifications: the traditional division into low (< 2,500 m) and high (> 2,500 m) altitudes, as well as the classification proposed by the International Society of Mountain Medicine, which delineates low (< 1,500 m), moderate (1,500–2,500 m), high (2,500–3,500 m), and very high (3,500–5,800 m) altitudes.

### Findings

From 2011–2021, we analyzed 49,765 IHD-related hospital admissions and 62,620 deaths. Men had an age-adjusted incidence rate of 55.08/100,000 and a mortality rate of 47.2/100,000, compared to 20.77/100,000 and 34.8/100,000 in women. Incidence and mortality surged in 2020 by 83% in men and 75% in women. Altitudinal stratification revealed higher IHD rates at lower altitudes (<2500 m), averaging 61.65 and 121.8 per 100,000 for incidence and mortality, which declined to 25.9 and 38.5 at elevations >2500 m. Men had more

download and can be accessed through the following link: https://github.com/covid19ec/DataStroke2 This includes all relevant datasets, materials, and code used in the study, allowing for independent verification and further analysis by other researchers.

**Funding:** The author(s) received no specific funding for this work.

**Competing interests:** The authors have declared that no competing interests exist.

pronounced rates across altitudes, exhibiting 138.7% and 150.0% higher incidence at low and high altitudes respectively, and mortality rates increased by 48.3% at low altitudes and 23.2% at high altitudes relative to women.

## Conclusion

Ecuador bears a significant burden of ischemic heart disease (IHD), with men being more affected than women in terms of incidence. However, women have a higher percentage of mortality post-hospital admission. Regarding elevation, our analysis, using two different altitude cutoff points, reveals higher mortality rates in low-altitude regions compared to high-altitude areas, suggesting a potential protective effect of high elevation on IHD risk. Nevertheless, a definitive dose-response relationship between high altitude and reduced IHD risk could not be conclusively established.

## 1. Introduction

Cardiovascular diseases (CVDs) are the leading cause of premature death and disability globally. These include ischemic heart disease (IHD), also known as coronary artery disease (CAD), and atherosclerotic cardiovascular disease (ACD) [1]. According to the World Health Organization (WHO), ischemic heart disease (IHD) was responsible for over 17.9 million deaths in 2019. In the same year, IHD accounted for more than 180 million disability-adjusted life-years (DALYs) and led to the loss of over 175 million years of life due to premature mortality [2, 3]. In Europe, ischemic heart disease (IHD) continues to be the most significant cause of death, accounting for over 860,000 deaths in men and nearly 880,000 deaths in women each year. In Brazil, the IHD landscape has evolved dramatically over the years, with the number of affected individuals soaring from 1.48 million in 1990 to over 4 million in 2019 [4–6]. In Ecuador, data reveals that between 2001 and 2016, there were 46,133 IHD-related deaths, with men constituting 60% of these fatalities [7]. Moreover, IHD was the leading cause of death in Ecuador in 2019 and 2020, and it ranked second in 2021 and 2022, only outpaced by COVID-19 [8, 9].

In terms of etiology, myocardial oxygen supply can be compromised by factors leading to hypoxia, anoxia, or infarction [10]. While traditional risk factors like smoking, obesity, and diabetes predominantly contribute to atherosclerosis in coronary arteries, non-traditional risk factors such as hypoxia and exposure to high altitudes remain relatively understudied [2, 11–13]. Given that millions globally reside at altitudes exceeding 2,500 meters in geographically diverse regions like the South American Andes, Central Asia's Karakorum-Pamir and Himalayan ranges, and the Ethiopian Highlands in East Africa, exploring high-altitude exposure as a risk factor for ischemic heart disease (IHD) is essential [14]. Theoretically, patients with coronary artery disease (CAD) and angina may be more adversely affected by high-altitude (HA) exposure due to pre-existing factors such as increased basal coronary flow, impaired arterial elasticity due to atheromatous lesions, and microvascular dysfunction [15–18]. Conversely, in healthy individuals, high-altitude exposure may induce coronary arterial vasodilation and increased blood flow, angiogenesis and therefore could potentially enhance resilience to hypoxic events [19–21].

In recent years, emerging evidence has enhanced our understanding of the interplay between ischemic heart disease (IHD) and high-altitude exposure, particularly concerning asymptomatic IHD patients and those who have undergone revascularization [18]. However, the question remains as to whether high altitude serves as a protective factor or increases the risk of infarction, especially in newcomers [22, 23].

This study aims to examine acute myocardial infarction and other ischemic heart diseases in Ecuador, a unique country with 221 cities spanning elevations from sea level to over 4,300 meters. The study serves dual purposes: it assesses the impact of ischemic heart disease within Ecuador and contributes to our understanding of the disease's burden at varying altitudes. By doing so, it adds a critical piece to the puzzle of understanding the relationship between elevation and cardiac disease.

## 2. Materials and methods

### 2.1 Study design

This is a nationwide, epidemiological, ecological study that examines the geographical distribution of ischemic heart disease in Ecuador from 2011 to 2021. We used hospital admissions as a proxy for incidence and death certificate data for mortality rates. The study includes all hospital admissions and deaths related to ischemic heart disease, categorized by the patient's place of residence as reported to the National Institute of Census and Statistics (INEC). This manuscript has been prepared in accordance with the STROBE guidelines, as detailed in the S1 Checklist.

### 2.2 Settings

Ecuador is a country located in the southern portion of the American continent covering approximately 283,560 km2 of land area. The Ecuadorian territory is divided into four geo-climatic regions: Coast, Andes (Highland), Amazonia and Galapagos Islands. It is also divided politically into 24 provinces, each of which is divided politically into territories called "cantons". Ecuador currently has a total of 221 cantons, 141 located at low altitude (<1,500 m), 28 at moderate altitude (1,500–2,500 m), 41 at high altitude (2,500–3,500 m) and 11 at very high altitude (3,500–5,500 m).

### 2.3 Population

According to official data (INEC), in 2017 Ecuador had a population of 17,082,730 inhabitants with a slight predominance of women (51%). According to ethnic distribution, 79.3% of population is mestizo (mixed and Spanish with native indigenous), 7.2% Afro-descendants, 7.1% indigenous, 6.1% white (Caucasian descendants) and 0.4% from other groups [24]. Regarding altitude, 60% of the population resides at low altitude, 10% at moderate altitude, 27% at high altitude and 3% at very high altitude [25].

### 2.4 Exposure

The association between altitude exposure with ischemic heart disease incidence and mortality was analyzed. The classification of low altitude <2,500 m and high altitude >2,500 m was used as a cut-off point for elevation exposure. While the classification offered by the International Society of Mountain Medicine: low altitude (<1,500 m), moderate altitude (1,500–2,500 m), high altitude (2,500–3,500 m) and very high altitude (3,500–5,500 m) was used to assess prevalence odds ratios by different elevations [26].

### 2.5 Outcome

Ischemic heart disease age, sex and altitude adjusted incidence and mortality rates were calculated using the total number of ischemic heart disease hospital admissions and all the cardiovascular diseases-related deaths in Ecuador from 2011 to 2021.

## 2.6 Data source

All cases corresponding to ICD-10 codes I21 (Acute Myocardial Infarction), I22 (Subsequent STEMI and NSTEMI), I23 (Current Complications Following STEMI and NSTEMI), I24 (Other Acute Ischemic Heart Diseases), and I25 (Chronic Ischemic Heart Disease) were extracted from the National Institute of Statistics and Census (INEC) [27]. The dataset comprised all registered cases of ischemic heart disease across Ecuador from 2011 to 2021.

## 2.7 Inclusion criteria

Using the 10th Revision of the International Classification of Diseases (ICD-10) the following subtypes of all ischemic heart disease cases and deaths were included: I21 (acute myocardial infarction), I22 (Subsequent myocardial infarction), I23 (Certain current complications following acute myocardial infarction), I24 (Other acute ischemic heart diseases), and I25 (Chronic ischemic heart diseases) and the combination of all of them in a new category called "all ischemic heart disease".

## 2.8 Exclusion criteria

Excluded from the analysis were cases with diagnoses other than ischemic heart disease as per ICD-10 codes I21, I22, I23, I24, and I25. Specifically, patients diagnosed with malignant cardiac conditions, valvular pathologies such as stenosis or insufficiencies, and congenital heart diseases were omitted from the study.

## 2.9 Bias

To reduce the possibility of incurring in some degree of selection bias and due to the nature of the data, three researchers (EOP, JIC, and RFN) downloaded the dataset and ran the analyses independently. To ensure that the data pertained to persons residing at different altitudes, the variable "place of residence" was used instead of the variable "place of medical attention".

## 2.10 Statistical analysis

In this comprehensive, epidemiological study, we aimed to scrutinize the incidence, mortality, and case-fatality rates of ischemic heart disease (IHD) in Ecuador from 2011 to 2021. Utilizing data from each province and canton, our analysis adjusted the primary metrics for sex and age based on the 2010–2021 Census data. Key variables included sex, age, month, and year of hospital admission. Both incidence and mortality rates were standardized for age and sex and expressed per 100,000 inhabitants. Hospital admissions served as a proxy for incidence, whereas death certificates informed mortality rates. Cases were classified across 17 age groups for nuanced interpretation. As an additional measure of the disease burden across varying altitudes, we calculated Years of Life Lost (YLL), using life expectancy data from the Ecuadorian National Institute of Statistics and Censuses. To investigate the effect of altitude on IHD metrics, we applied Poisson regression models, adjusted for age and sex. Join point regression was utilized to identify significant temporal changes in incidence and mortality. Statistical tests for comparing subgroups included T-tests for two categories and ANOVA for more than two categories.

## 2.11 Ethical consideration

The dataset employed in this study was procured from the publicly accessible databases maintained by the National Institute of Statistics and Census (INEC) of Ecuador, which exclusively consist of anonymized, non-identifiable information. This ensures the preservation of both

ethical integrity and confidentiality regarding the subjects involved. Compliant with international Good Clinical Practice (GCP) guidelines and conforming to the ethical principles delineated in the Declaration of Helsinki, the use of such anonymized data sets is deemed ethically acceptable, as they safeguard individuals from harm and maintain stringent confidentiality standards. Importantly, this study sought and received formal exemption from ethical review by the Institutional Review Board (IRB) of the Ethics Committee at Universidad de las Américas (UDLA), Quito, Ecuador. The project was officially assigned the code 2023-EXC-008 and the exemption was formally granted on May 8, 2023.

## 3. Results

### 3.1 Sex and age analysis

During the study period (2011–2021), 49,765 hospital admissions and 62,620 deaths were recorded due to ischemic heart disease (IHD) using the ICD codes: I21, I22, I23, I24, and I25. Men had a higher average age of 64.08 (±13.14) years compared to women patients who had an average age of 67.79 (±13.98) years. Men had a higher proportion of ischemic heart disease, accounting for 71.4% (n = 35,549), while women accounted for 28.6% (n = 14,216), representing a 62.5% lower incidence rate compared to men. The overall distribution of mortality rates varied significantly by age groups. For instance, the younger populations (0 to 49 years) accounted the 11.52% of the total number of cases, registered, nevertheless, their average mortality was under 4.8% of the cases. The association between mortality and age was assessed using Pearson's correlation analysis. For men, the correlation coefficient (r) was 0.016, falling within a 95% confidence interval of [-0.454; 0.479]. Similarly, for women, the correlation coefficient was calculated as 0.07, with a 95% confidence interval of [-0.41; 0.52]. Much like the findings in men, this coefficient suggests a very weak or non-existent linear relationship between the variables.

### 3.2 Incidence

During the analyzed period (2011–2021), the overall sex-specific adjusted incidence rate of ischemic heart disease was higher in men, with 55.08/100,000 compared to 20.77/100,000 in women. The age groups most affected were patients between 75 and 79 years old and those over 80 years of age, with the highest incidence rates observed in both sexes. Furthermore, 90% of women patients and 88% of men patients admitted to the hospital for ischemic heart disease were older than 50 years of age (Table 1).

From 2011 onwards, the annual incidence of IHD, estimated using hospital admissions as a proxy, has been on average 39.3 per 100,000 for men and 15.5 per 100,000 for women (S1 Checklist). Notably, in 2020, during the onset and first waves of the COVID-19 pandemic, the incidence rate of cardiac ischemia showed a sharp decrease, reaching its lowest level in recent years at 31.6 per 100,000 for men and 12.1 per 100,000 for women (Fig 1).

**Table 1. Displays the incidence and mortality rates of ischemic heart disease in Ecuador from 2011 to 2021 according to different age categories and gender.**

| Age Group | Age Range | Cases (n) | Incidence/100,000 | Mortality/100,000 | Cases (n) | Incidence/100,000 | Mortality/100,000 |
|---|---|---|---|---|---|---|---|
| | | | **Women** | | | **Men** | |
| **Childhood & Adolescence** | 0–19 | 56 | 0.4 | 0.3 | 85 | 0.5 | 0.6 |
| **Early Adulthood** | 20–39 | 394 | 1.5 | 2.2 | 1,346 | 5.3 | 6.9 |
| **Midlife** | 40–59 | 3,235 | 19.6 | 17.9 | 10,796 | 70.7 | 43.2 |
| **Senior** | 60+ | 10,531 | 124.2 | 362.8 | 23,322 | 290.3 | 499.6 |
| **Total** | | 14,216 | 36.2 | 95.5 | 35,555 | 91.7 | 137.6 |

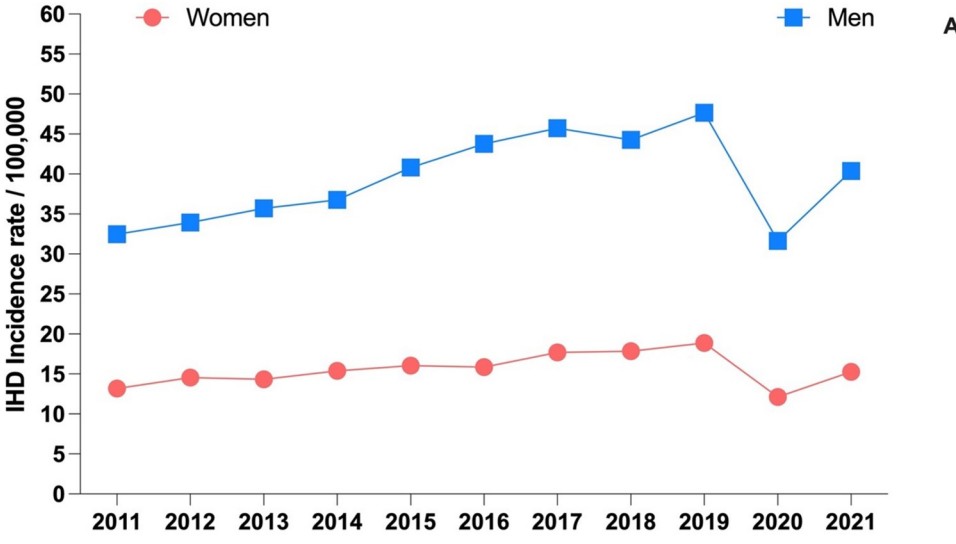

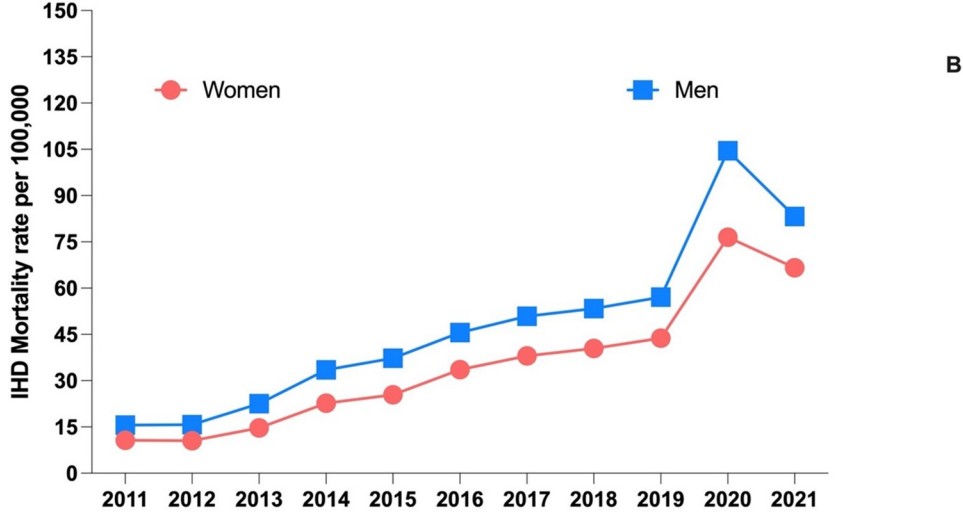

**Fig 1.** A illustrates the gender-specific trend in incidence rate of ischemic heart disease from 2011 to 2021. Men consistently had a higher incidence rate than women during this period, with a slight decrease observed for both genders in 2020. B shows the mortality rate of ischemic heart disease, with a significant increase in 2020–2021 compared to previous years. Men had a consistently higher mortality rate than women.

### 3.3 Mortality

From 2011 to 2021, the mortality rate for ischemic heart disease, based on death certificates, was 47.2 per 100,000 for men and 34.8 per 100,000 for women on average. However, in 2020 there was a significant increase of approximately 121.4% in men (with a mortality rate of 104.5 per 100,000) and approximately 119.8% in women (with a mortality rate of 76.5 per 100,000). When considering in-hospital mortality, the average percentage of patients with ischemic heart disease who died was 8.7% for men and 13.8% for women. Acute myocardial infarction had the highest in-hospital mortality rate for both sexes, with a rate of 13% for men and 21.7% for women (Fig 2).

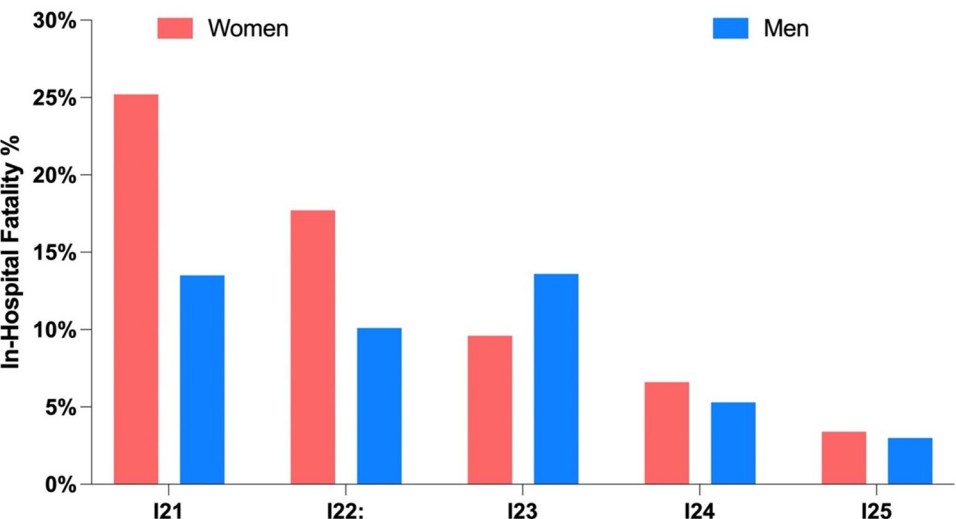

**Fig 2. Shows the in-hospital mortality rate (%) by cause for men and women from 2011 to 2021.** Except for certain current complications following acute myocardial infarction, women had higher mortality rates than men for all causes.

### 3.4 Time trends and joint point regression analyses

In the study, joint regression analyses were conducted to identify breakpoints in IHD incidence and mortality rates, considering multiple covariates such as age, gender, province and elevation using cantonal data. For incidence rate with respect to gender, a significant breakpoint was identified in the year 2016, indicating a change in trend. Notably, male gender was positively associated with an increased incidence rate ($p < 0.001$). The model had a strong fit, with an adjusted R-squared value of 0.8563. For mortality rate, a breakpoint was observed in 2012. Male gender was also significantly associated with an increased mortality rate ($p < 0.001$), and the model demonstrated a robust adjusted R-squared value of 0.8916. When province was considered as a covariate for incidence rate, a breakpoint was identified in 2020. Several provinces showed statistically significant associations, and the model had an adjusted R-squared of 0.6254. These findings suggest temporal shifts and demographic variables that warrant focused interventions.

### 3.5 Geodemographic distribution

**3.5.1 Province distribution.** The incidence rates of ischemic heart disease were highest in the Galapagos (167.7 / 100,000), Manabí (75.48 / 100,000), and Pastaza (72.43 / 100,000) provinces, while the lowest rates were found in Cotopaxi, Esmeraldas, and Chimborazo provinces with 26.68 / 100,000, 27.29 / 100,000, and 34.15 / 100,000, respectively (Table 2). In terms of mortality, Galapagos and Guayas provinces had the highest mortality rates with 180.7/100,000 and 92.47/100,000, respectively (Table 2).

**3.5.2 Cantonal distribution.** According to the patient's recorded place of residence, the cantons in Ecuador with the highest incidence rates per 100,000 population for IHD were Aguarico, Paquisha, and Yacuambi with rates of 2,552.26, 1,970.58, and 1,782.82, respectively. On the other hand, the cantons with the lowest incidence rates were Quininde, Otavalo, and Mejia with rates of 37.88, 38.59, and 43.26, respectively.

In terms of mortality rates per 100,000 population, the cantons with the highest rates for IHD were Exterior, Crnel. Marcelino Mariduena, and Olemdo with "Inf" indicating an

**Table 2. Incidence and mortality rates for ischemic heart disease in Ecuadorian provinces from 2011 to 2021, by province.** The provinces with the highest incidence rates for ischemic heart disease in Ecuador from highest to lowest are Galapagos, Manabí, and Cotopaxi. The provinces with the highest mortality rates for ischemic heart disease in Ecuador from highest to lowest are Orellana, Pastaza, and Galapagos.

| Province | Cases n | C. Incidence Rate | A. Incidence Rate | Deaths n | C. Mortality Rate | A. Mortality Rate |
|---|---|---|---|---|---|---|
| **Azuay** | 2482 | 46.9 [46.39;47.4] | 47.27 [46.76;47.78] | 2536 | 48.75 [47.57;49.93] | 51.52 [50.21;52.82] |
| **Bolivar** | 290 | 44.2 [43.64;44.76] | 44.85 [44.25;45.44] | 703 | 82.97 [80.53;85.41] | 86.8 [84.16;89.43] |
| **Cañar** | 485 | 52.08 [51.48;52.68] | 52.49 [51.89;53.1] | 843 | 80.54 [77.86;83.23] | 84.32 [81.49;87.15] |
| **Carchi** | 274 | 46 [45.46;46.53] | 46.21 [45.68;46.75] | 496 | 72.12 [70.08;74.17] | 72.34 [70.33;74.34] |
| **Chimborazo** | 821 | 34.15 [33.76;34.55] | 34.56 [34.16;34.95] | 1761 | 60.3 [58.75;61.86] | 62.69 [61.03;64.35] |
| **Cotopaxi** | 551 | 26.68 [26.39;26.97] | 27.11 [26.81;27.42] | 1153 | 47.58 [46.06;49.11] | 48.85 [47.29;50.41] |
| **El Oro** | 1947 | 44.42 [43.95;44.9] | 42.48 [42.04;42.92] | 1918 | 42.33 [41.5;43.16] | 42.62 [41.73;43.51] |
| **Esmeraldas** | 895 | 27.29 [27.01;27.56] | 26.75 [26.48;27.01] | 1364 | 46.77 [45.74;47.8] | 46.71 [45.62;47.79] |
| **Galapagos** | 96 | 167.7 [164.46;170.94] | 164.36 [161.21;167.51] | 75 | 180.7 [176.82;184.57] | 180.37 [176.83;183.92] |
| **Guayas** | 15155 | 44.73 [44.3;45.15] | 42.32 [41.93;42.7] | 31789 | 92.47 [90.53;94.41] | 91.58 [89.49;93.67] |
| **Imbabura** | 794 | 39.07 [38.62;39.51] | 39.55 [39.09;40.01] | 1424 | 57.09 [55.75;58.42] | 59.32 [57.89;60.76] |
| **Loja** | 1366 | 43.83 [43.31;44.35] | 43.85 [43.33;44.37] | 1340 | 50.59 [49.19;52] | 52.34 [50.83;53.85] |
| **Los Rios** | 2064 | 39.82 [39.37;40.27] | 39.35 [38.88;39.81] | 5386 | 87.22 [85.5;88.95] | 88.5 [86.63;90.38] |
| **Manabi** | 8120 | 75.48 [74.78;76.19] | 73.39 [72.71;74.07] | 8103 | 77.33 [75.6;79.05] | 79.52 [77.58;81.45] |
| **Morona Santiago** | 280 | 48.91 [48.37;49.45] | 49.17 [48.62;49.72] | 195 | 55.99 [54.58;57.39] | 58.41 [56.85;59.97] |
| **Napo** | 130 | 53.31 [52.51;54.1] | 52.41 [51.62;53.19] | 106 | 47.43 [46.13;48.72] | 47.49 [46.19;48.79] |
| **Orellana** | 148 | 50.86 [50.19;51.53] | 49.98 [49.29;50.68] | 103 | 36.61 [35.97;37.26] | 34.33 [33.81;34.85] |
| **Pastaza** | 141 | 72.43 [71.52;73.34] | 72.78 [71.9;73.66] | 137 | 88.42 [86.35;90.49] | 93.26 [90.9;95.62] |
| **Pichincha** | 9775 | 41.96 [41.59;42.34] | 39.88 [39.52;40.23] | 10626 | 42.99 [41.99;43.98] | 42 [41;43.01] |
| **Santa Elena** | 723 | 44.36 [43.9;44.82] | 43.33 [42.89;43.78] | 1348 | 70.92 [69.18;72.66] | 70.05 [68.28;71.82] |
| **Santo Domingo** | 1061 | 51.32 [50.83;51.8] | 49.28 [48.84;49.73] | 1737 | 70.75 [69.42;72.08] | 68.61 [67.28;69.95] |
| **Sucumbios** | 266 | 37.38 [36.98;37.78] | 36.14 [35.75;36.52] | 212 | 36.5 [35.87;37.13] | 34.96 [34.36;35.57] |
| **Tungurahua** | 1716 | 51.3 [50.73;51.87] | 52.69 [52.06;53.33] | 2393 | 66.8 [65.08;68.51] | 70.78 [68.87;72.69] |
| **Zamora Chinchipe** | 152 | 66.62 [65.63;67.62] | 67.27 [66.23;68.31] | 79 | 48.66 [47.61;49.72] | 51.95 [50.69;53.21] |

C: crude, A: adjusted

undefined value. In contrast, the cantons with the lowest mortality rates were Quito, Cuenca, and Tena with rates of 46.82, 52.72, and 59.18, respectively (Fig 3).

### 3.6 Altitude analysis

The analysis of ischemic heart disease incidence and mortality rates in relation to altitude reveals distinct patterns. At low altitudes (<2500 m), the average incidence rate is 61.65, and the average mortality rate is 121.8. In contrast, at high altitudes (>2500 m), the average incidence rate drops to 25.9, and the average mortality rate decreases to 38.5 (Fig 4). These findings suggest that both the incidence and mortality rates of ischemic heart disease are significantly higher at lower altitudes compared to higher altitudes (Fig 4).

**3.6.1 Age and sex differences at different altitudes.** The analysis of ischemic heart disease incidence and mortality rates in relation to altitude and gender reveals considerable differences. At both low and high altitudes, men have substantially higher incidence rates compared to women, with a percentage difference of 138.7% at low altitudes and 150.0% at high altitudes. Mortality rates also exhibit disparities, although less pronounced, with men showing a 48.3% higher rate at low altitudes and a 23.2% higher rate at high altitudes compared to women (Table 3).

As expected, the incidence of the disease increases with age for both men and women at any altitude. The highest rates are observed in the 80+ age group, particularly in the low

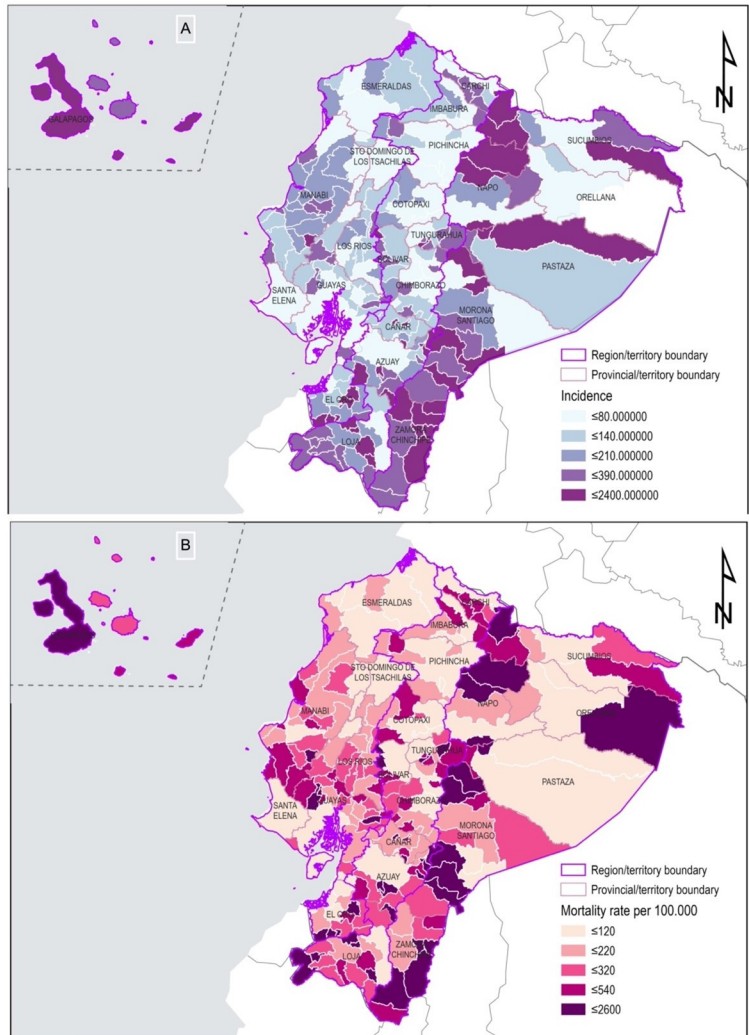

**Fig 3. Spatial analysis of ischemic heart disease in Ecuador at the cantonal level (2011–2021).** A) Incidence rate per 100,000 population, illustrating the distribution of ischemic heart disease across 221 cantons. B) Mortality rate per 100,000 population, revealing the patterns of fatal outcomes across the same 221 cantons.

altitude range, with men experiencing a rate of 936.11 per 100,000 individuals and women showing a rate of 789.63 per 100,000 individuals. This trend of higher rates among men compared to women is consistent across all age groups and altitude ranges. The percentage difference between very high altitude compared to low altitude is 97.71% for men and 97.11% for women. This significant difference indicates that the incidence of ischemic heart disease is much higher in individuals living at low altitudes compared to those living at very high altitudes (Fig 5).

### 3.7 Burden of diseases at different altitudes

Our investigation into the burden of ischemic heart disease (I21, I22, I23, I24, I25) at different altitudes revealed a higher burden in low altitude compared to high altitude areas for both men and women. In terms of years of life lost (YLL), the percentage difference between low and high-altitude areas was 251.07% for men and 219.57% for women. When considering YLL

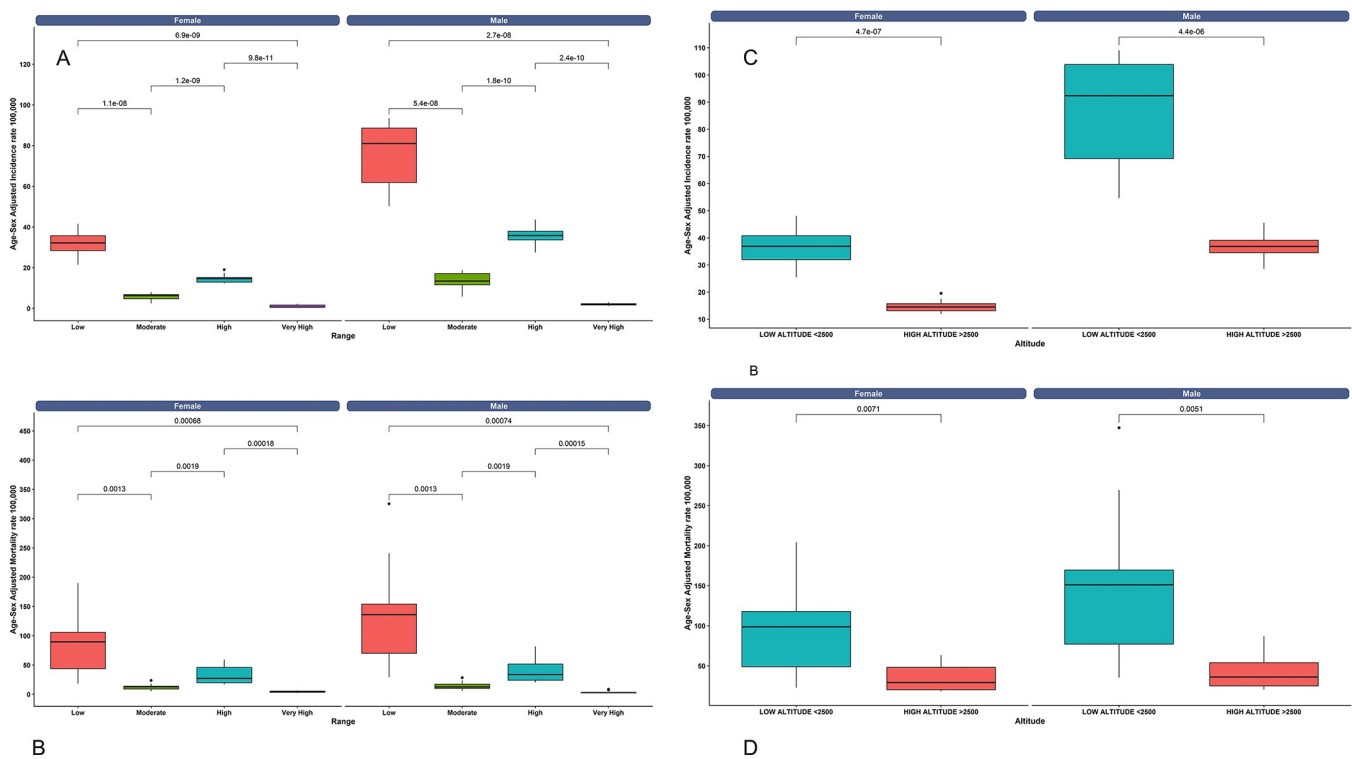

**Fig 4.** A) Incidence rate of ischemic heart disease per 100,000 population across the four altitude categories: low (<1,500 m), moderate (1,500 m to 2,500 m), high (2,500 m to 3,500 m), and very high (3,500 m to 5,500 m), B) Mortality rate of ischemic heart disease per 100,000 population in men and women categorized in low (<1,500 m), moderate (1,500 m to 2,500 m), high (2,500 m to 3,500 m), and very high (3,500 m to 5,500 m). C) Incidence rate categorized by low (<2,500 m) and high (>2,500 m) altitude. D) Mortality rate of ischemic heart disease per 100,000 population in men and women, categorized by low (<2,500 m) and high (>2,500 m) altitude.

**Table 3. Ischemic heart disease incidence and mortality rates according to different altitudes from 2011 to 2021.**

|  |  | Cases |  |  | Deaths |  |
|---|---|---|---|---|---|---|
| Altitude range | n | Adjusted Incidence Rate |  | n | Adjusted Mortality Rate |  |
| Short Classification |  | Men | Women |  | Men | Women |
| Low altitude <2500 m | 36060 | 86,9 [60,1–107,1] | 36,4 [25,9–45,6] | 58328 | 145,5 {37,2–308,3] | 98,1 [24,4–202,2] |
| High altitude >2500 m | 13672 | 37,0 [29,9–44,5] | 14,8 [12,2–18,5] | 17336 | 42,5 [21,1–80,2] | 34,5 [18,2–61,8] |
| No information | 33 | N/A | N/A | 52 | N/A | N/A |
| **Total** | 49765 | N/A | N/A | 75716 | N/A | N/A |
| ISMM Classification |  |  |  |  |  |  |
| Low (0–1500 m) | 31399 | 62,4 [0,42–195,1] | 26,7 [0,12–100,7] | 52970 | 117,7 [0,48–487,2] | 71,1 [0,14–283,4] |
| Moderate (1500–2500 m) | 4661 | 11,0 [0,13–38,6] | 4,12 [0,16–15,9] | 5358 | 12,2 [0,13–51,4] | 9,74 [0,17–44,0] |
| High (2500–3500 m) | 13321 | 31,1 [0,21–100,6] | 12,1[0,15–45,3] | 16352 | 34,5 [0,14–146,6] | 27,7 [0,2–121,9] |
| Very High (3500–5800 m) | 351 | 1,23 [0,13–4,7] | 0,66 [0,14–1,94] | 984 | 2,6 [0,13–11,6] | 2,42 [0,15–11,1] |
| No information | 33 | N/A | N/A | 52 | N/A | N/A |
| **Total** | 49765 | N/A | N/A | 75716 | N/A | N/A |

m: meters above sea level, ISMM: International Society of Mountain Medicine.

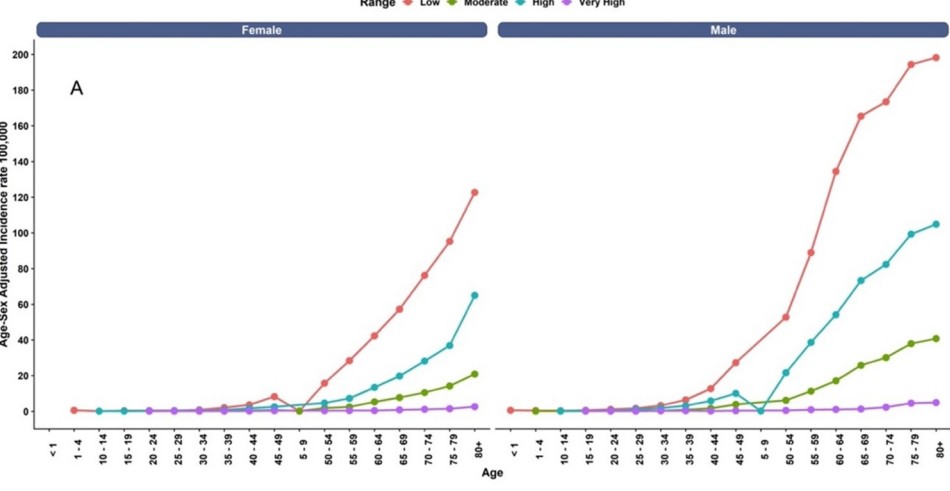

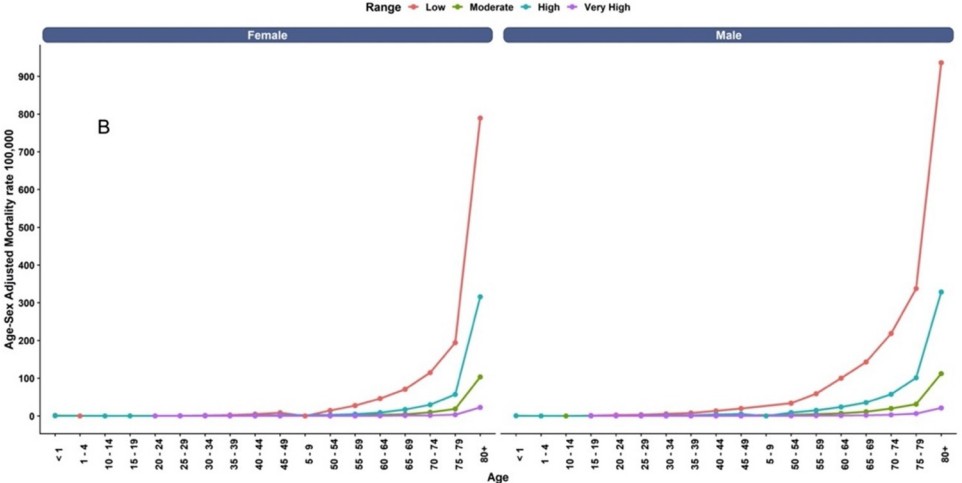

**Fig 5.** A) Age-specific incidence rates of ischemic heart disease per 100,000 population, stratified by sex and altitude categories (low, moderate, high, and very high). B) Age-specific mortality rates of ischemic heart disease per 100,000 population.

per 100,000 people, the percentage difference was 236.96% for men and 207.24% for women. These findings demonstrate a significantly higher burden of ischemic heart disease in low altitude areas, with men experiencing a greater impact than women across both altitude categories.

## 4. Discussion

In this study, we have presented one of the very first comprehensive analysis of the burden of ischemic heart disease across distinct elevation ranges using countrywide data. The initial observation that drew our attention was the decline in incidence during the first year of the pandemic in Ecuador. Both men and women experienced reduced access to healthcare systems, which likely contributed to the reduced hospital admission but the sharp increase in IHD related-mortality [28]. This increase was inversely proportional to hospitalization rates,

suggesting that a higher number of individuals fell ill, but due to limited access to hospitals, more people died without receiving medical care [29]. Furthermore, this trend could be associated with the surge in excess mortality observed during the COVID-19 pandemic [30]. Related to high altitude exposure, the country's diverse topography, with 221 cities situated from sea level up to 4,300 meters (very high altitudes), offers a unique opportunity to investigate the impact of altitude on the epidemiology of chronic diseases such as ischemic heart disease (IHD) or stroke [31, 32], for example, Freire et al. stated that the main reasons for Galapagos being one of the provinces of Ecuador with the highest rates of overweight and obesity is the increased consumption of ultra-processed foods and decreased consumption of traditional cooking based in unprocessed and minimally processed foods, mainly due to the greater ease of access to this type of food by the inhabitants of the islands [33], their findings could justify, at least partially, the higher rates of coronary heart disease triggered by metabolic alterations in this population [34].

The assessment of the geographical distribution of ischemic heart disease across Ecuador provides critical insights into the prevalence and impact of this condition in a setting where it has been under-researched. While our study highlights distinct epidemiological variations in incidence and mortality rates between populations at lower and higher altitudes, it is crucial to consider several confounding variables. Factors such as poverty, limited healthcare access, dietary habits, and smoking may influence these rates. Despite recognizing these limitations, it is important to note that obtaining more comprehensive data in Ecuador is particularly challenging due to data unavailability or restricted access. Nonetheless, we discovered that the provinces of Ecuador with the highest incidence of ischemic heart disease were Galápagos, Manabí, and Pastaza. These provinces are located at low altitudes, where it seems that certain risk factors may have a stronger influence on the development of both ischemic heart disease and stroke [31, 35]. Interestingly, a study carried out in the Swiss population suggest potential explanations for the lower mortality from coronary heart disease (CHD) and stroke at higher altitudes. These authors suggest that some behavioral cardiovascular diseases (CVD) risk factors, such as smoking, poor diet, and physical inactivity, could be more common at lower altitudes, although the same study advises that these risk factors do not differ substantially by altitude [20, 35].

In Ecuador, epidemiological data reveals key risk factors for ischemic heart disease (IHD). Specifically, 10% of the population aged 18–69 are tobacco users, while 60% face issues related to overweight or obesity, most prominently between the ages of 40 and 50. Galapagos province exhibits the highest obesity rate at 15.09% [36–38]. Moreover, 19.8% of Ecuadorians are hypertensive; among them, 17% have uncontrolled hypertension and 56.3% are without medication. Pan American Health Organization data further indicate that 10% of individuals aged 50–59 suffer from diabetes, and 50% have elevated cholesterol levels. Importantly, these studies lack controls for altitude and canton of residence, variables that may influence the prevalence of these risk factors [39, 40].

The disparate mortality rates between high-altitude and low-altitude residents remain not fully elucidated. However, lifestyle and environmental factors unique to higher altitudes could contribute. Recent research suggests that conditions predisposing to hypoxia, such as sleep apnea, may confer cardioprotective effects through mechanisms like myocardial angiogenesis [41–44]. In this context, we hypothesize that the reduced partial pressure of oxygen at higher altitudes may confer protective cardiovascular effects. This hypothesis is informed by animal studies demonstrating that chronic hypoxia induces angiogenesis, thereby enhancing the brain's resilience to ischemic hypoxia [19, 21, 42]. In fact, intermittent exposure to hypobaric hypoxia has been proposed, and successfully applied, for the improvement of myocardial perfusion both in people and in experimental animal models [44, 45]. Other research posits that

residents of high-altitude regions may adopt healthier lifestyles, such as regular physical exercise and improved dietary habits, possibly influenced by the region's demanding climate and geography. These lifestyle factors could contribute to enhanced cardiovascular health [46, 47]. Other factors, such as increased exposure to ultraviolet radiation and higher levels of vitamin D synthesis at higher altitudes, may also play a protective role against cardiovascular disease, but more investigation is needed [48]. This liposoluble vitamine exerts cardio-protective effects by modulating blood pressure through its action on endothelial and smooth muscle cells [48]. A deficiency in vitamin D has been implicated in vascular dysfunction, increased arterial stiffness, and left ventricular hypertrophy, according to current scientific literature [48–50].

Despite uncertainties in the underlying mechanisms that might reduce the risk of ischemic heart disease (IHD), our study clearly demonstrates a higher prevalence of IHD and associated mortality at lower altitudes, after adjustments for population, age, and sex. These results align with findings from other studies conducted in high-altitude locations. For instance, Lopez-Pascual et al., who studied an Ecuadorian population and found lower prevalence rates of metabolic syndrome, hypercholesterolemia, and hyperglycemia among adults with a high educational level living at 2,754m compared to inhabitants of the Ecuadorian lowland coast region [51]. These results together with those reported by Freire et al. [33], suggest that lifestyle and environmental factors associated with living at low altitudes, such as dietary patterns, sedentary but stressful urban life or different habits and access to healthcare, may contribute to the higher mortality rates observed in our study [51]. Other authors have compared this phenomenon in other populations, finding a greater relationship of hypercholesterolemia with socio-economic and sociocultural level and lifestyles rather than with altitude [52].

While our study found that mortality rates caused by cardiovascular disease are generally lower at higher altitudes, it is important to note that at altitudes greater than 3,500m, stroke mortality rates increase again and may surpass those observed in moderate-lying areas [31]. This finding suggests that physiological responses to permanent life at high altitude may have their limits (at least in Andean people) and due to extremely high blood viscosity could have negative effects on tissue oxygenation. Other studies have also pointed out the relationship between extreme altitudes above 3,000m and increased mortality in general, which may be due to factors such as decreased oxygen availability and increased exposure to extreme weather conditions [52].

Our study, focusing on a wide range of altitudes including regions at extreme elevations, complements existing literature which mostly reports cardiovascular data from moderate altitudes [52]. Altitude-related genetic and metabolic adaptations may influence cardiovascular health, but their mechanisms remain incompletely understood [53]. While studies suggest that altitude-specific behaviors may affect metabolism and energy efficiency [51]. the role of factors like hypoxia-induced angiogenesis or hemoglobin levels varies across populations and can even differ within high-altitude communities [53–55].

According to our findings, we cannot assert a consistent dose-response relationship between high altitude, although a reduced risk of developing ischemic heart disease seems apparent between 2,000m and 3,500m of elevation. However, beyond this altitude range, other factors might be involved in an increase of harm effect.

## 5. Limitations

Our research is primarily based on INEC's publicly accessible databases, bringing with it inherent limitations related to data accuracy and completeness. We were not able to verify the

initial data, rendering our results susceptible to biases stemming from data entry errors or incomplete reporting.

One significant constraint is the absence of control for key risk factors such as body mass index, blood pressure levels, smoking status, alcohol consumption, diabetes, a sedentary life-style, cholesterol, and triglyceride levels. This is due to the unavailability of these data points. The inability to adjust for these factors inherently limits the precision of our risk estimations.

Another limitation is that our dataset does not provide the granularity needed to identify individual patients who might have utilized healthcare services in multiple locations at different times, potentially resulting in duplicated data entries.

We also lacked the necessary information to employ a proportional hazards model for survival rate estimations, which limits the depth of our statistical analyses.

Our study's ecological design further restricts our ability to establish causal relationships between variables.

Additionally, the regional limitations of our data set mean that our findings may not be generalizable to other geographic areas or populations, especially those with significantly different healthcare infrastructures or cultural attitudes toward healthcare.

Moreover, the study did not consider seasonal variations, which might influence ischemic heart disease incidence and mortality, nor did it account for healthcare quality, which could have differential impacts at various altitudes.

Lastly, although our study identifies correlations between altitude and ischemic heart disease, it does not delve into underlying biological mechanisms, leaving this as a subject for future research.

By acknowledging these limitations, we aim to adhere to academic rigor and offer transparent insight into the potential scope and constraints of our study. This also provides a roadmap for future research in this area.

## 6. Conclusions

Our investigation sheds light on the intricate relationship between altitude and mortality attributable to ischemic heart disease (IHD). While our data suggest that elevated altitudes may exert a protective effect against IHD, the relationship between altitude and cardiovascular health appears to be multifaceted and potentially dependent on specific altitude ranges. Although a consistent dose-response relationship was not observed in our study, it is imperative that future research explores additional variables, including dietary habits, alcohol and tobacco use, and physical activity levels, which could influence cardiovascular risk.

Our findings hold particular relevance for high-altitude populations in the Andean regions, offering insights that could guide targeted interventions to alleviate disease burden. In summary, our research underscores the critical need for further studies to comprehensively understand the effects of altitude on cardiovascular health, as well as the interactions with various modifiable and non-modifiable risk factors.

## Supporting information

**S1 Checklist. STROBE guidelines: Provides the STROBE guidelines used in this study, ensuring detailed and transparent reporting.**
(PDF)

**S1 Table. Shows incidence and mortality rates of ischemic heart disease in Ecuador (2011–2021) by age and gender.**
(PDF)

## Acknowledgments

We express our gratitude to the National Institute of Statistics and Census for making their data publicly available and facilitating our research endeavors.

## Author Contributions

**Conceptualization:** Esteban Ortiz-Prado.

**Data curation:** Esteban Ortiz-Prado, Juan S. Izquierdo-Condoy, Raúl Fernández-Naranjo.

**Formal analysis:** Esteban Ortiz-Prado, Juan S. Izquierdo-Condoy, Raúl Fernández-Naranjo, Erick Duque, Maria Gabriela Davila Rosero.

**Funding acquisition:** Esteban Ortiz-Prado.

**Investigation:** Esteban Ortiz-Prado, Juan S. Izquierdo-Condoy, Jorge Vásconez-González, Leonardo Cano, Ana Carolina González, Estefanía Morales-Lapo, Galo S. Guerrero-Castillo, Erick Duque, Maria Gabriela Davila Rosero, Diego Egas, Ginés Viscor.

**Methodology:** Esteban Ortiz-Prado, Juan S. Izquierdo-Condoy, Estefanía Morales-Lapo, Diego Egas, Ginés Viscor.

**Project administration:** Raúl Fernández-Naranjo, Jorge Vásconez-González.

**Supervision:** Esteban Ortiz-Prado, Diego Egas, Ginés Viscor.

**Validation:** Esteban Ortiz-Prado, Juan S. Izquierdo-Condoy, Raúl Fernández-Naranjo, Jorge Vásconez-González, Leonardo Cano, Ana Carolina González, Estefanía Morales-Lapo, Galo S. Guerrero-Castillo, Erick Duque, Maria Gabriela Davila Rosero, Diego Egas, Ginés Viscor.

**Visualization:** Esteban Ortiz-Prado, Raúl Fernández-Naranjo, Maria Gabriela Davila Rosero, Diego Egas, Ginés Viscor.

**Writing – original draft:** Esteban Ortiz-Prado, Juan S. Izquierdo-Condoy, Jorge Vásconez-González, Leonardo Cano, Ana Carolina González, Estefanía Morales-Lapo, Galo S. Guerrero-Castillo, Erick Duque, Maria Gabriela Davila Rosero.

**Writing – review & editing:** Esteban Ortiz-Prado, Juan S. Izquierdo-Condoy, Jorge Vásconez-González, Leonardo Cano, Ana Carolina González, Estefanía Morales-Lapo, Diego Egas, Ginés Viscor.

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
