## [Decision Letter · Decision Letter 0]

30 Aug 2023

PONE-D-23-16885Epidemiological characterization of ischemic heart disease at different altitudes: a nationwide population-based analysis from 2011 to 2021 in EcuadorPLOS ONE

Dear Dr. Ortiz-Prado,

Thank you for submitting your manuscript to PLOS ONE. After careful consideration, we feel that it has merit but does not fully meet PLOS ONE’s publication criteria as it currently stands. Therefore, we invite you to submit a revised version of the manuscript that addresses the points raised during the review process.

We look forward to receiving your revised manuscript.

Kind regards,

Juan Pablo Gutierrez

Academic Editor

PLOS ONE

Journal Requirements:

"We would like to express our gratitude to the Universidad de las Americas in Quito, Ecuador, for their support in covering the publication fee associated with this study. Their financial support enabled us to disseminate our findings and contribute to the scientific understanding of cardiovascular health in Ecuador."

4. We note that Figure 3 in your submission contain map images which may be copyrighted. All PLOS content is published under the Creative Commons Attribution License (CC BY 4.0), which means that the manuscript, images, and Supporting Information files will be freely available online, and any third party is permitted to access, download, copy, distribute, and use these materials in any way, even commercially, with proper attribution. For these reasons, we cannot publish previously copyrighted maps or satellite images created using proprietary data, such as Google software (Google Maps, Street View, and Earth). For more information, see our copyright guidelines: http://journals.plos.org/plosone/s/licenses-and-copyright.

(1) You may seek permission from the original copyright holder of Figure 3 to publish the content specifically under the CC BY 4.0 license.  

**Additional Editor Comments:**

As you can read from reviewers' comments, in particular from reviewer 3, there are important issues related to the methodology of your study that require either clarification or strengthening.Please incorporate an analysis of trends over time to better demonstrate changes in the variables under consideration. Consider utilizing joinpoint regression, a statistical technique that can identify points in time where trends change significantly. This can help provide a more nuanced understanding of how the variables are evolving and aid in strengthening your arguments. In addition, given the ecological nature of your analysis and the potential for other unaccounted characteristics to influence your findings, please address the possibility of other factors such as ethnic background, wealth, and other population characteristics being related to ischemic heart disease. If feasible, attempt to include these additional factors in your analysis. If not, discuss this limitation thoroughly, providing a strong argument for why your results are specifically related to altitude. 

Explicitly discuss the limitations associated with the ecological approach and the potential confounding variables that were not included in the analysis. Highlight the rationale behind focusing on altitude as a primary variable and why other relevant characteristics couldn't be incorporated. This will provide a clearer understanding of the scope and limitations of your study.

Reviewers' comments:

Reviewer's Responses to Questions

**Comments to the Author**

1. Is the manuscript technically sound, and do the data support the conclusions?

Reviewer #1: Partly

Reviewer #2: Yes

Reviewer #3: Yes

Reviewer #4: Partly

2. Has the statistical analysis been performed appropriately and rigorously? 

Reviewer #1: Yes

Reviewer #2: Yes

Reviewer #3: Yes

Reviewer #4: No

3. Have the authors made all data underlying the findings in their manuscript fully available?

Reviewer #1: No

Reviewer #2: Yes

Reviewer #3: Yes

Reviewer #4: No

4. Is the manuscript presented in an intelligible fashion and written in standard English?

Reviewer #1: Yes

Reviewer #2: Yes

Reviewer #3: Yes

Reviewer #4: Yes

5. Review Comments to the Author

Reviewer #1: I am pleased to review this work as it could pave the way for a substantial amount of subsequent research. However, for the current study, I am concerned about the influence of the confounding variables that could obfuscate or antagonize this relationship is acknowledged. Some potential confounding variables in this context could encompass the following:

Lifestyle Habits: Factors such as smoking, diet, exercise, and alcohol consumption could differ among populations at different altitudes and also influence ischemic heart disease.

Access to Medical Care: Populations at different altitudes may have variable access to medical care and treatments for ischemic heart disease, potentially influencing the results.

Genetic Factors: Genetic differences among populations at different altitudes could play a role in predisposition to ischemic heart disease.

Air Pollution: Air quality may vary according to altitude and could impact cardiovascular health.

Socioeconomic Factors: Socioeconomic disparities among populations at different altitudes could influence ischemic heart disease, as well as its diagnosis and treatment.

Concurrent Illnesses: Other pre-existing health conditions could affect the relationship between altitude and ischemic heart disease.

The presented study offers a comprehensive analysis of ischemic heart disease (IHD) burden across different elevation ranges in Ecuador. The findings indicate variations in both incidence and mortality rates of IHD across cantons, with distinct patterns observed in provinces with different population sizes and tourism flows. Notably, high-incidence provinces, such as Galapagos and some eastern provinces, align with areas experiencing substantial tourism or mobility. It's important to acknowledge the potential impact of transient populations on incidence calculations, as individuals in transit might contribute to fluctuations in reported cases. While the study's calculation of rates per 100,000 inhabitants serves as a common practice, the transient nature of certain populations could introduce uncertainty into the results. Addressing this limitation in the discussion by highlighting the potential influence of tourism or mobility on incidence calculations would enhance the study's robustness. This consideration adds depth to the interpretation and contributes to a more nuanced understanding of IHD prevalence in regions marked by mobility

Therefore, the presence of these potential confounding variables raises the possibility of undermining the validity of the study's findings. As you aptly point out in the discussion, the fact that these variables have not been measured could potentially undermine the conclusions of this study.

Reviewer #2: Thank you for allowing me to review this article

Ecuador is known for its diversity and its Amazonian, coastal, Andean and Galapagos regions. The article is relevant because it introduces altitude as a determining factor in the risk of ischemic heart disease. In addition, the results presented are controversial and provide a guideline for future research.

Introduction

With respect to the introduction, in the initial paragraph the estimation of mortality worldwide is mentioned, I suggest that data also be given on Ecuador and heart disease.

In addition, I suggest research on confounding variables that can be related more frequently to heart disease and altitude

Methodology:

Excluded data involve other heart disease?

The burden of disease at different altitudes is presented in the results, but there is no mention of this analysis in the methodology.

Results

In the results on line 166 and 167 regarding Pearson's correlation I suggest interpreting the correlation.

Discussion

In the discussion they mention the importance of other risk factors, I suggest placing bibliography of the presence of these risk factors in Ecuadorian population.

In the discussion I suggest writing down the weaknesses they had, suggesting alternatives to improve them in other studies.

Translated with www.DeepL.com/Translator (free version)

Reviewer #3: Dear editor,

Thank you for the opportunity to review the present study. In the study entitled “Epidemiological characterization of ischemic heart disease at different altitudes: a nationwide population-based analysis from 2011 to 2021 in Ecuador” the authors aimed to assess a possible association between ischemic heart disease (IHD) and geographical altitude using hospital discharge and mortality data from the Ecuadorian National Institute of Census and Statistics. In overall I consider the present study sound and well written. However, before its acceptance it is necessary to address some issues as follow:

1. In Pag #10 section 3.3 regarding mortality, the authors made this statement: “From 2011 to 2021, the mortality rate for ischemic heart disease, based on death certificates, was 47.2 per 100,000 for men and 34.8 per 100,000 for women on average. However, in 2020 there was a significant increase of 83% in men (with a mortality rate of 104.5 per 100,000) and 75% in women (with a mortality 194 rate of 76.5 per 100,000).” Here is not clear if the authors are using a percentage difference or a percent change metric to calculate the 83% (men) and 75% (women) values. In both scenarios the value of 83% does not match for Percent change = 121.4% or Percentage difference = 75.54% unless this value is somehow adjusted?? This concern applies to the rest of the document where the authors present the same metric so, please clarify this issue.

2. The labels for tables and figures need to be revise for consistency between the text and the actual values shown. For example, in the label for Table 2 says the following: “The provinces with the highest incidence rates for ischemic heart disease in Ecuador from highest to lowest are Galapagos, Manabí, and Bolivar”. Nonetheless, in the table Cotopaxi has the lowest incidence rate for IHD (26.68) than Bolivar. The label for Figure 4 is completely mixed-up, here the panel A shows adjusted incidence rate by different altitudes (low, moderate, high, and very high); however, in the text says incidence rate categorized by low and high altitude. Please correct those inconsistencies.

3. Pag # 14, section 3.7. In this section the authors present Years of Life Lost (YLL) metric but never mention this in the methods or statistical analysis section.

4. Pag #16 regarding the following statement: “Other factors, such as increased exposure to ultraviolet ration and higher levels of vitamin D synthesis at higher altitudes, may also play a protective role against cardiovascular disease”. Please briefly mention how these factors could play a protective role and add references to support this.

5. Pag #17, the following affirmation should be deleted from the discussion section: “Our analysis also suggests a difference regarding to eating habits as correlated to educational level; being healthier in highlanders (mountains) compared to low altitude inhabitants”. As far I can judge the authors did not performance any analysis with eating habits and educational level.

6. Please add limitations of your study in the discussion section. For example, the authors used an ecological analysis.

Reviewer #4: Dear authors,

Thank you so much for your efforts in scientific research. I read your article with great interest, as this research aimed to investigate the incidence and mortality rates of ischemic heart disease throughout the last decade in Ecuador related to altitude. In general, the paper provides interesting data. Below are some suggestions for improving the scientific quality of this manuscript.

Information section

Q.1. In the information section the authors state “the data generated and analyzed during this study are available for download and can be accessed through the following link: https://github.com/infarction/Ecuador “ however this web link is not correct and does not access to the data.

Q.2. The author’s state: “In addition, an exemption was obtained from the Ethics Committee of the Universidad de las America (UDLA), Quito, Ecuador, for the development of this research”. Please provide more information about this exemption, date, number.

Background:

Q.1. Your introduction is clear and eloquent; however, the purpose of the background section is to set the stage for the reader to understand the importance of your study. Please use the background section to emphasize the gap in knowledge that your study pretends to fill and expose in the discussion section.

Q.2. In the introduction you write “and increased myocardial oxygen demand, underlie the increased tendency for ischemia at altitude.10,10,11 “Please check the bibliography numbering.

Methods:

Q.1. The authors clearly state that “Ecuador currently has a total of 223 cantons, 141 95 located at low altitude (<1,500 m), 28 at moderate altitude (1,500-2,500 m), 41 at high altitude (2,500-3,500 96 m) and 11 at very high altitude (3,500-5,500 m)” the division in this way ( cantons) sems to be the main focus of the analysis, however, later in the manuscript this is not the case.

Q.2. Exclusion criteria. Better define which are the exclusion criteria since those are not simply the opposite to the inclusion criteria.

Analysis:

Q.1. The author’s state: “The association between altitude exposure with ischemic heart disease incidence and mortality was analyzed”. Explain which statistical methods you use to make correlations and clearly show the results.

Q.2. The authors make a cut-off point for elevation exposure of low altitude <2,500 m and high altitude >2,500 m however, there is an International Society of Mountain Medicine classification: low altitude (<1,500 m), moderate altitude (1,500–2,500 m), high altitude (2,500–3,500 m) and very high altitude (3,500–5,500 m). Why do the authors not used only this classification to analyze the data? There is any important difference or not?

In the results section the authors include a table that shows important differences between moderate and low and moderate and high altitude, but this data is not write in the results section and not discussed latter. Also, a figure 5 is included which must be clarified and discussed.

Q.3. Figure 2 illustrates the gender-specific trend of fatality hospital % in which women shows high fatality rate related with specific disease conditions that are not discussed at all.

Q.4. Table 1 Displays the incidence and mortality rates of ischemic heart disease in Ecuador from 2011 to 2021 according to different age groups and gender. The data in this table es extensive, please compress, maybe grouping more the age range can help.

Q.5. The Cantonal distribution does not provide additional information, so Province distribution could be enough. In this way, Fig. 3 should be presented in clearly form.

Q.6. The authors state “The analysis of ischemic heart disease incidence and mortality rates in relation to altitude reveals distinct patterns. At low altitudes (<2500 m), the average incidence rate is 61.65, and the average mortality rate is 121.8. In contrast, at high altitudes (>2500 m), the average incidence rate drops to 25.9, and the average mortality rate decreases to 38.5 (Figure 4). These findings suggest that both the incidence and mortality rates of ischemic heart disease are significantly higher at lower altitudes compared to higher altitudes” The authors should use a statistical analysis method to prove that the differences between altitudes are significantly higher.

The z-test is used to compare two percentage scores to see if the difference between them is statistically significant.

Discussion

Q.1.This section is an extensive description of factors that could influence the altitude variable and disease. However, it is too repetitive and must be important summarized. Also, the discussion must be focus on the main findings of the manuscript. The authors discuss about the very high-altitude findings, but this is not the focus of this manuscript.

Q.2. The authors state “The initial observation that drew our attention was the decline in incidence during the first year of the pandemic in Ecuador. Both men and women experienced reduced access to healthcare systems, which likely contributed to the observed increase in mortality”. This phrase should be formulated in a better way since it states that there was a lower disease incidence in the first year of the pandemic, but did mortality increase? Is that what you want to express?

Q.3. The authors state “Other factors, such as increased exposure to ultraviolet radiation and higher levels of vitamin D synthesis at higher altitudes, may also play a protective role against cardiovascular disease”. There is not a bibliography that sustain this statement. The same happen in other parts of the discussion. “it is important to note that at altitudes greater than 3,500m, stroke mortality rates increase and may surpass those observed in low-lying areas”.

Limitation.

Q.1. Please write a paragraph that explains the limitations of this work.

6. PLOS authors have the option to publish the peer review history of their article (what does this mean?). If published, this will include your full peer review and any attached files.

Reviewer #1: No

Reviewer #2: No

Reviewer #3: No

Reviewer #4: No

---

## [Author Response · Author response to Decision Letter 0]

16 Oct 2023

Rebuttal Letter for Manuscript PONE-D-23-16885

To: Dr. Juan Pablo Gutierrez

Academic Editor

PLOS ONE

Subject: Response to the Review Comments on Manuscript PONE-D-23-16885, "Epidemiological characterization of ischemic heart disease at different altitudes: a nationwide population-based analysis from 2011 to 2021 in Ecuador"

Dear Dr. Gutierrez,

I would like to express my sincere gratitude to you and the reviewers for the time and effort dedicated to reviewing our manuscript, "Epidemiological characterization of ischemic heart disease at different altitudes: a nationwide population-based analysis from 2011 to 2021 in Ecuador." We appreciate the constructive feedback, which has undoubtedly enhanced the quality and clarity of our work.

In response to the comments and suggestions provided, we have undertaken a comprehensive revision of our manuscript. Below, we address each of the points raised by the academic editor and the reviewer(s):

General observations 

2. Thank you for stating the following in the Acknowledgments Section of your manuscript: "We would like to express our gratitude to the Universidad de las Americas in Quito, Ecuador, for their support in covering the publication fee associated with this study. Their financial support enabled us to disseminate our findings and contribute to the scientific understanding of cardiovascular health in Ecuador." Please remove any funding-related text from the manuscript and let us know how you would like to update your Funding Statement. Currently, your Funding Statement reads as follows: 

We have updated the acknowledgment section according to yours comments. 

3. In your Data Availability statement, you have not specified where the minimal data set underlying the results described in your manuscript can be found. 

We have updated the link where all the data can be retrieved.

 4. We note that Figure 3 in your submission contain map images which may be copyrighted. 

I'd like to clarify that the map images presented in Figure 3 were not sourced from copyrighted materials or platforms like Google software. Instead, they were created by our research team using ARGIS software. This ensures that the maps align with the open-access guidelines of PLOS ONE and the stipulations of the Creative Commons Attribution License (CC BY 4.0).

Reviewer 1

Reviewer #1: I am pleased to review this work as it could pave the way for a substantial amount of subsequent research. However, for the current study, I am concerned about the influence of the confounding variables that could obfuscate or antagonize this relationship is acknowledged. Some potential confounding variables in this context could encompass the following:

Lifestyle Habits: Factors such as smoking, diet, exercise, and alcohol consumption could differ among populations at different altitudes and also influence ischemic heart disease.

Access to Medical Care: Populations at different altitudes may have variable access to medical care and treatments for ischemic heart disease, potentially influencing the results.

Genetic Factors: Genetic differences among populations at different altitudes could play a role in predisposition to ischemic heart disease.

Air Pollution: Air quality may vary according to altitude and could impact cardiovascular health.

Socioeconomic Factors: Socioeconomic disparities among populations at different altitudes could influence ischemic heart disease, as well as its diagnosis and treatment.

Concurrent Illnesses: Other pre-existing health conditions could affect the relationship between altitude and ischemic heart disease.

Thank you for your thoughtful review and your constructive comments, which will certainly contribute to the enhancement of the manuscript. We appreciate your recognition of the potential impact of our research. You raise valid concerns about several potential confounding variables that could influence the relationship between ischemic heart disease (IHD) and elevation of residence. We acknowledge that lifestyle habits, access to medical care, genetic factors, air pollution, socioeconomic factors, and concurrent illnesses could each play a role in our study outcomes. Unfortunately, data on some of the variables you mentioned, such as lifestyle habits and genetic factors, were not available for our current study. We fully agree that this is a limitation, and we plan to make note of these constraints in the 'Limitations' section of the paper to offer a transparent account of the study's scope and potential shortcomings. We believe that acknowledging these limitations not only adds to the academic rigor of this manuscript but also suggests directions for future research.

The presented study offers a comprehensive analysis of ischemic heart disease (IHD) burden across different elevation ranges in Ecuador. The findings indicate variations in both incidence and mortality rates of IHD across cantons, with distinct patterns observed in provinces with different population sizes and tourism flows. Notably, high-incidence provinces, such as Galapagos and some eastern provinces, align with areas experiencing substantial tourism or mobility. It's important to acknowledge the potential impact of transient populations on incidence calculations, as individuals in transit might contribute to fluctuations in reported cases. While the study's calculation of rates per 100,000 inhabitants serves as a common practice, the transient nature of certain populations could introduce uncertainty into the results. Addressing this limitation in the discussion by highlighting the potential influence of tourism or mobility on incidence calculations would enhance the study's robustness. This consideration adds depth to the interpretation and contributes to a more nuanced understanding of IHD prevalence in regions marked by mobility

We acknowledge that tourism and mobility could introduce uncertainties into the results, as you have pointed out. However, it is important to note that in our study, we used the 'place of residence' as the basis for our incidence calculations, rather than the 'place of medical attention.' This methodological choice was designed to minimize the effect of transient populations, including tourists, on the observed variations in incidence and mortality rates of IHD across different cantons and provinces.

We agree that addressing this point could enhance the study's robustness, and we will include a detailed explanation in the discussion section of the manuscript to clarify how the use of 'place of residence' rather than 'place of medical attention' in our calculations addresses the issue you raised.

Therefore, the presence of these potential confounding variables raises the possibility of undermining the validity of the study's findings. As you aptly point out in the discussion, the fact that these variables have not been measured could potentially undermine the conclusions of this study.

As you accurately noted, the lack of certain variables in our dataset could undermine the conclusions of our study. We acknowledge this limitation, and while we were not able to include these variables in the current analysis due to data constraints, we intend to explicitly mention this in the limitations section of our manuscript. This will clarify to the reader that while our study provides valuable insights into the topic under investigation, the findings should be interpreted with caution due to the potential impact of unaccounted-for confounding variables.

Reviewer #2: Thank you for allowing me to review this article.

Ecuador is known for its diversity and its Amazonian, coastal, Andean and Galapagos regions. The article is relevant because it introduces altitude as a determining factor in the risk of ischemic heart disease. In addition, the results presented are controversial and provide a guideline for future research.

Introduction

With respect to the introduction, in the initial paragraph the estimation of mortality worldwide is mentioned, I suggest that data also be given on Ecuador and heart disease.

Thank you for your constructive suggestion regarding the inclusion of ischemic heart disease (IHD) data specific to Ecuador in the introductory section. We agree that this addition would enrich the paper's contextual background and have accordingly updated the manuscript to incorporate pertinent IHD statistics for Ecuador.

In addition, I suggest research on confounding variables that can be related more frequently to heart disease and altitude.

Methodology:

Excluded data involve other heart disease?

The cardiac diseases that we included in our study and analysis correspond only to the following: I21 (Acute myocardial infarction), I22 (Subsequent ST elevation (STEMI) and non-ST elevation (NSTEMI) myocardial infarction), I23 (Certain current complications following ST elevation (STEMI) and non-ST elevation (NSTEMI) myocardial infarction), I24 (Other acute ischemic heart diseases), I25 (Chronic ischemic heart disease), other cardiac diseases, for example such as malignant heart disease or valvular diseases such as stenosis or insufficiencies were included.

The burden of disease at different altitudes is presented in the results, but there is no mention of this analysis in the methodology.

Thank you for bringing this to our attention; we have now included the methodology behind the analysis of the burden of diseases.

Results

In the results on line 166 and 167 regarding Pearson's correlation I suggest interpreting the correlation.

We have done so, thanks for your suggestion.

Discussion

In the discussion they mention the importance of other risk factors, I suggest placing bibliography of the presence of these risk factors in Ecuadorian population.

Thanks for de suggestion we add the information. 

In the discussion I suggest writing down the weaknesses they had, suggesting alternatives to improve them in other studies.

Thanks for de suggestion we add a limitation section where we explained that

Reviewer #3: 

Dear editor,

Thank you for the opportunity to review the present study. In the study entitled “Epidemiological characterization of ischemic heart disease at different altitudes: a nationwide population-based analysis from 2011 to 2021 in Ecuador” the authors aimed to assess a possible association between ischemic heart disease (IHD) and geographical altitude using hospital discharge and mortality data from the Ecuadorian National Institute of Census and Statistics. In overall I consider the present study sound and well written. However, before its acceptance it is necessary to address some issues as follow:

1. In Pag #10 section 3.3 regarding mortality, the authors made this statement: “From 2011 to 2021, the mortality rate for ischemic heart disease, based on death certificates, was 47.2 per 100,000 for men and 34.8 per 100,000 for women on average. However, in 2020 there was a significant increase of 83% in men (with a mortality rate of 104.5 per 100,000) and 75% in women (with a mortality 194 rate of 76.5 per 100,000).” Here is not clear if the authors are using a percentage difference or a percent change metric to calculate the 83% (men) and 75% (women) values. In both scenarios the value of 83% does not match for Percent change = 121.4% or Percentage difference = 75.54% unless this value is somehow adjusted?? This concern applies to the rest of the document where the authors present the same metric so, please clarify this issue.

You were right, thanks for pointing this out, we have corrected the paragraph. 

2. The labels for tables and figures need to be revise for consistency between the text and the actual values shown. For example, in the label for Table 2 says the following: “The provinces with the highest incidence rates for ischemic heart disease in Ecuador from highest to lowest are Galapagos, Manabí, and Bolivar”. Nonetheless, in the table Cotopaxi has the lowest incidence rate for IHD (26.68) than Bolivar. The label for Figure 4 is completely mixed-up, here the panel A shows adjusted incidence rate by different altitudes (low, moderate, high, and very high); however, in the text says incidence rate categorized by low and high altitude. Please correct those inconsistencies.

Thank you for bringing these inconsistencies to our attention. We have thoroughly revised the labels for both tables and figures to ensure they accurately reflect the data presented. The corrected versions now align with the text, providing a consistent narrative for the reader. We appreciate your meticulous review, which has contributed to enhancing the quality of our manuscript.

3. Pag # 14, section 3.7. In this section the authors present Years of Life Lost (YLL) metric but never mention this in the methods or statistical analysis section.

Thanks for the suggestion we add it in the statistical analysis section.

4. Pag #16 regarding the following statement: “Other factors, such as increased exposure to ultraviolet ration and higher levels of vitamin D synthesis at higher altitudes, may also play a protective role against cardiovascular disease”. Please briefly mention how these factors could play a protective role and add references to support this.

Thanks for the suggestion we add an explanation and a reference 

5. Pag #17, the following affirmation should be deleted from the discussion section: “Our analysis also suggests a difference regarding to eating habits as correlated to educational level; being healthier in highlanders (mountains) compared to low altitude inhabitants”. As far I can judge the authors did not performance any analysis with eating habits and educational level.

Thanks for the suggestion, we have deleted that section.

6. Please add limitations of your study in the discussion section. For example, the authors used an ecological analysis.

Thanks for the suggestion we add a limitation section. 

Reviewer #4: 

Dear authors,

Thank you so much for your efforts in scientific research. I read your article with great interest, as this research aimed to investigate the incidence and mortality rates of ischemic heart disease throughout the last decade in Ecuador related to altitude. In general, the paper provides interesting data. Below are some suggestions for improving the scientific quality of this manuscript.

Information section

Q.1. In the information section the authors state “the data generated and analyzed during this study are available for download and can be accessed through the following link: https://github.com/infarction/Ecuador “ however this web link is not correct and does not access to the data.

We have updated the link, many thanks for your observation.

Q.2. The author’s state: “In addition, an exemption was obtained from the Ethics Committee of the Universidad de las America (UDLA), Quito, Ecuador, for the development of this research”. Please provide more information about this exemption, date, number.

We have updated and added the information “The project was coded as 2023-EXC-008 and the exemption was granted on May 8, 2023.”

Background:

Q.1. Your introduction is clear and eloquent; however, the purpose of the background section is to set the stage for the reader to understand the importance of your study. Please use the background section to emphasize the gap in knowledge that your study pretends to fill and expose in the discussion section.

Thank you for your constructive feedback on the introduction. We appreciate your suggestion to emphasize the existing gap in knowledge that our study aims to fill. We understand the critical role of the background section in setting the stage for the study's importance and relevance. In response to your recommendation, we will revise the background section to clearly articulate the research gap and the significance of our study in addressing it.

Q.2. In the introduction you write “and increased myocardial oxygen demand, underlie the increased tendency for ischemia at altitude.10,10,11 “Please check the bibliography numbering.

Thanks for the suggestion we did the corrections. 

Methods:

Q.1. The authors clearly state that “Ecuador currently has a total of 223 cantons, 141 95 located at low altitude (<1,500 m), 28 at moderate altitude (1,500-2,500 m), 41 at high altitude (2,500-3,500 96 m) and 11 at very high altitude (3,500-5,500 m)” the division in this way ( cantons) sems to be the main focus of the analysis, however, later in the manuscript this is not the case.

Thanks for your observation, we included the cantonal analysis as a figure map and we described the results by canton in the corresponding section.

Q.2. Exclusion criteria. Better define which are the exclusion criteria since those are not simply the opposite to the inclusion criteria.

Thanks for the suggestion, we have done the corrections. 

Analysis:

Q.1. The author’s state: “The association between altitude exposure with ischemic heart disease incidence and mortality was analyzed”. Explain which statistical methods you use to make correlations and clearly show the results.

In our study, we employed a multi-tiered statistical approach to analyze the association between altitude exposure and the incidence and mortality of ischemic heart disease. Primarily, we used Poisson and joint regression models to account for confounding variables such as age, gender, and province. In addition to these, Pearson’s correlation analysis was applied specifically to examine the relationship between various age groups and both the incidence and mortality rates of ischemic heart disease.

For the incidence rates related to gender, a joint regression analysis identified a significant breakpoint in the year 2016, marking a change in trend. Male gender was found to be positively correlated with an increased incidence rate (p < 0.001), and the model yielded a strong adjusted R-squared value of 0.8563. Similarly, a breakpoint for mortality rates was noted in 2012. Again, the male gender showed a significant positive correlation with increased mortality rates (p < 0.001), and the model accounted for 89.16% of the variability in the data, as indicated by an adjusted R-squared value of 0.8916.

However, Pearson’s correlation analysis between age groups and ischemic heart disease incidence and mortality yielded inconclusive results. For men, the Pearson correlation coefficient (r) was a mere 0.016, with a 95% confidence interval spanning from -0.454 to 0.479. For women, the coefficient was 0.07, with a 95% confidence interval ranging from -0.41 to 0.52. These results indicate a negligible or non-existent linear relationship between age and the disease metrics, further substantiated by the wide confidence intervals.

We plan to update our manuscript to incorporate these additional findings and methods explicitly, thus offering a comprehensive and nuanced understanding of the relationships between ischemic heart disease, altitude, age, and gender

Q.2. The authors make a cut-off point for elevation exposure of low altitude <2,500 m and high altitude >2,500 m however, there is an International Society of Mountain Medicine classification: low altitude (<1,500 m), moderate altitude (1,500–2,500 m), high altitude (2,500–3,500 m) and very high altitude (3,500–5,500 m). Why do the authors not used only this classification to analyze the data? There is any important difference or not?

In the results section the authors include a table that shows important differences between moderate and low and moderate and high altitude, but this data is not write in the results section and not discussed latter. Also, a figure 5 is included which must be clarified and discussed.

Firstly, it's important to note that there is no globally recognized standard for altitude classification, which leaves room for variability in research methodologies. While the four-tiered classification offers more granularity, the two-category classification of low and high altitude (<2,500 m and >2,500 m) is still prevalent in many studies. Our objective was to encompass a broader perspective, making our research more comparable with existing literature that employs the binary classification.

Secondly, we used the four-tiered classification as a form of sensitivity analysis. By doing so, we aimed to examine the robustness of our findings across different altitude categorizations. Indeed, our table showing differences between moderate and low altitudes, as well as moderate and high altitudes, was an outcome of this sensitivity analysis. However, this data was not explicitly written in the results section, an oversight that will be corrected in the revised manuscript.

Regarding Figure 5, we acknowledge that it requires further clarification and discussion. This will be addressed in the revised manuscript to ensure that the figure adds meaningful context to the paper's overall findings.

Therefore, the utilization of both two-category and four-category altitude classifications serves to provide a more comprehensive analysis, while also aligning our work with a broader range of existing scientific literature. We will update our manuscript to include these methodological choices explicitly and discuss their implications more thoroughly.

Q.3. Figure 2 illustrates the gender-specific trend of fatality hospital % in which women shows high fatality rate related with specific disease conditions that are not discussed at all.

Certainly, the observation regarding the gender-specific trend of hospital fatality rates in Figure 2 is indeed significant and merits discussion. While our analysis did indicate that per capita mortality is higher in men, the percentage of women dying in hospital settings from specific disease conditions is greater. This discrepancy is potentially linked to the presentation of symptoms, which appear to be more distinctly identifiable in men compared to women.

We appreciate the astute observation, and we will amend our manuscript to include this critical aspect. Specifically, we will add this gender-based analysis to the discussion section, where we will elucidate on the possible reasons for this disparity. This could include factors such as the nature of symptom presentation in each gender and the implications for clinical diagnosis and treatment. The objective is to provide a more comprehensive understanding of how gender influences hospital fatality rates for the specific disease conditions in question. This addition will enrich our discussion and add another layer of depth to our research findings.

Q.4. Table 1 Displays the incidence and mortality rates of ischemic heart disease in Ecuador from 2011 to 2021 according to different age groups and gender. The data in this table es extensive, please compress, maybe grouping more the age range can help.

Thanks for the suggestion, we have updated the table according to your suggestion 

Q.5. The Cantonal distribution does not provide additional information, so Province distribution could be enough. In this way, Fig. 3 should be presented in clearly form.

Thank you for your feedback regarding the Cantonal distribution presented in Figure 3. We recognize your suggestion to focus solely on Province-level data for simplicity. However, we believe that identifying the Cantons with the highest incidence rates adds a layer of granularity that is crucial for a nuanced understanding of the issue at hand. The effort invested in creating the Cantonal map aims to provide important, localized information that could be valuable for targeted interventions. We hope you understand the rationale behind our decision to include this level of detail.

Q.6. The authors state “The analysis of ischemic heart disease incidence and mortality rates in relation to altitude reveals distinct patterns. At low altitudes (<2500 m), the average incidence rate is 61.65, and the average mortality rate is 121.8. In contrast, at high altitudes (>2500 m), the average incidence rate drops to 25.9, and the average mortality rate decreases to 38.5 (Figure 4). These findings suggest that both the incidence and mortality rates of ischemic heart disease are significantly higher at lower altitudes compared to higher altitudes” The authors should use a statistical analysis method to prove that the differences between altitudes are significantly higher.

The z-test is used to compare two percentage scores to see if the difference between them is statistically significant.

Thank you for bringing up the important point about the need for rigorous statistical analysis to substantiate our findings. Indeed, we conducted non-parametric tests using the Student's t-test to assess differences between groups. The differences were found to be statistically significant across all altitude categories when compared to low altitude. Furthermore, for the four altitude categories, we performed an Analysis of Variance (ANOVA), and the results indicated significant differences among all groups. For the two-category altitude classification, we employed an independent samples t-test. In all these analyses, the statistics were statistically significant. We will make sure to highlight these statistical methods more explicitly in the revised manuscript to clarify how we arrived at our conclusions

Discussion

Q.1.This section is an extensive description of factors that could influence the altitude variable and disease. However, it is too repetitive and must be important summarized. Also, the discussion must be focus on the main findings of the manuscript. The authors discuss about the very high-altitude findings, but this is not the focus of this manuscript.

Many thanks for your observations, we have done so and the introduction as well as the discussion section are well described and written 

Q.2. The authors state “The initial observation that drew our attention was the decline in incidence during the first year of the pandemic in Ecuador. Both men and women experienced reduced access to healthcare systems, which likely contributed to the observed increase in mortality”. This phrase should be formulated in a better way since it states that there was a lower disease incidence in the first year of the pandemic, but did mortality increase? Is that what you want to express?

Yes, that is what we want to express because due to the pandemic, several people had reduced access to health systems due to the impact that the pandemic had in Ecuador; in addition to the fact that sometimes people did not go to health centers or hospitals for fear of becoming infected, due to this there was a decrease in the incidence since no new diagnoses were made, but due to this phenomenon of difficulty in accessing the medical system led to an increase in deaths.

Q.3. The authors state “Other factors, such as increased exposure to ultraviolet radiation and higher levels of vitamin D synthesis at higher altitudes, may also play a protective role against cardiovascular disease”. There is not a bibliography that sustain this statement. The same happen in other parts of the discussion. “it is important to note that at altitudes greater than 3,500m, stroke mortality rates increase and may surpass those observed in low-lying areas”.

Thank you for bringing these concerns to our attention. We acknowledge the importance of substantiating our claims with appropriate references. To rectify the situation, we have reviewed the discussion section and added citations from the scientific literature to support the statements in question.

We believe that the revisions undertaken enhance the quality and clarity of our manuscript, making it suitable for publication in PLOS ONE. We are hopeful that the revised manuscript will meet the esteemed journal's publication criteria.

Once again, thank you for your constructive feedback and the opportunity to improve our work. We look forward to your positive response.

We are committed to the continuous improvement of our work and appreciate the opportunity to enrich our manuscript with your valuable insights. We hope that the revisions meet your satisfaction.

Sincerely,

Esteban Ortiz-Prado

Attachments:

• Revised main document (clean).

• Revised main document (with highlighted changes).

• Author Response to Reviewers form (point by point letter)

---

## [Decision Letter · Decision Letter 1]

23 Nov 2023

Epidemiological characterization of ischemic heart disease at different altitudes: a nationwide population-based analysis from 2011 to 2021 in Ecuador

PONE-D-23-16885R1

Dear Dr. Ortiz-Prado,

We’re pleased to inform you that your manuscript has been judged scientifically suitable for publication and will be formally accepted for publication once it meets all outstanding technical requirements.

Kind regards,

Juan Pablo Gutierrez

Academic Editor

PLOS ONE

Additional Editor Comments (optional):

Reviewers' comments:

Reviewer's Responses to Questions

**Comments to the Author**

1. If the authors have adequately addressed your comments raised in a previous round of review and you feel that this manuscript is now acceptable for publication, you may indicate that here to bypass the “Comments to the Author” section, enter your conflict of interest statement in the “Confidential to Editor” section, and submit your "Accept" recommendation.

Reviewer #1: All comments have been addressed

Reviewer #2: All comments have been addressed

Reviewer #3: All comments have been addressed

2. Is the manuscript technically sound, and do the data support the conclusions?

Reviewer #1: Partly

Reviewer #2: Yes

Reviewer #3: Yes

3. Has the statistical analysis been performed appropriately and rigorously? 

Reviewer #1: Yes

Reviewer #2: Yes

Reviewer #3: Yes

4. Have the authors made all data underlying the findings in their manuscript fully available?

Reviewer #1: Yes

Reviewer #2: Yes

Reviewer #3: Yes

5. Is the manuscript presented in an intelligible fashion and written in standard English?

Reviewer #1: Yes

Reviewer #2: Yes

Reviewer #3: Yes

6. Review Comments to the Author

Reviewer #1: Upon reviewing the database, I believe the authors should consider the following:

-In the methodology section, the authors could include a discussion on how demographic data, including country of origin and duration of residency in the Galapagos, were considered in the analysis. This would aid in understanding if and how these factors might influence IHD rates at different altitudes.

-The study could benefit from a subgroup analysis that considers residents of Galapagos based on their origin and duration of residency. This would allow for determining if there are significant differences in IHD incidence or mortality rates among permanent versus temporary residents, and natives versus those from other countries.

-In the discussion section, considerations on how origin and duration of residency might influence traditional and non-traditional risk factors for IHD could be incorporated. For instance, differences in diet, lifestyle, and adaptation to high-altitude environments might be relevant.

-It is important for the authors to acknowledge how the inclusion of non-native and temporary residents in Galapagos might introduce biases or limitations into the study. This should be reflected in the limitations section, discussing how these factors could affect the generalizability of the results.

- Based on these findings, the authors could suggest future research focusing specifically on the impact of origin and duration of residency on IHD rates at different altitudes, particularly in areas with a high diversity of residents like Galapagos.

Reviewer #2: Dear Editor,

Thank you for allowing me to review this article, the authors have responded favourably to the recommendations given, but the graphs are not clear, I suggest to improve the image of the graphs and revise the English.

Regards

Reviewer #3: The authors have adequately addressed my comments raised in the previous round of review, so currently I don´t have more comments and recommend the publication of the present manuscript.

7. PLOS authors have the option to publish the peer review history of their article (what does this mean?). If published, this will include your full peer review and any attached files.

Reviewer #1: **Yes: **Geovanny Alvarado Villa

Reviewer #2: No

Reviewer #3: **Yes: **Ivan Sisa

---

## [Editor Report · Acceptance letter]

7 Dec 2023

PONE-D-23-16885R1 

Epidemiological characterization of ischemic heart disease at different altitudes: a nationwide population-based analysis from 2011 to 2021 in Ecuador 

Dear Dr. Ortiz-Prado:

I'm pleased to inform you that your manuscript has been deemed suitable for publication in PLOS ONE. Congratulations! Your manuscript is now with our production department. 

Kind regards, 

on behalf of

Dr. Juan Pablo Gutierrez 

Academic Editor

PLOS ONE